# Cellular Localization of Carbonic Anhydrase Nce103p in *Candida albicans* and *Candida parapsilosis*

**DOI:** 10.3390/ijms21030850

**Published:** 2020-01-28

**Authors:** Jiří Dostál, Jan Blaha, Romana Hadravová, Martin Hubálek, Olga Heidingsfeld, Iva Pichová

**Affiliations:** 1Institute of Organic Chemistry and Biochemistry of the Czech Academy of Sciences, Flemingovo náměstí 2, 166 10 Prague, Czech Republic; jiri.dostal@uochb.cas.cz (J.D.); jan.blaha.18@ucl.ac.uk (J.B.); hadravova@uochb.cas.cz (R.H.); hubalek@uochb.cas.cz (M.H.); olga-hh@uochb.cas.cz (O.H.); 2Department of Biochemistry, Faculty of Science, Charles University in Prague, Hlavova 2030, 128 43 Prague, Czech Republic

**Keywords:** carbonic anhydrase, Nce103p, *Candida albicans*, Candida parapsilosis, localization, cell wall, electron microscopy, mass spectrometry

## Abstract

Pathogenic yeasts *Candida albicans* and *Candida parapsilosis* possess a ß-type carbonic anhydrase Nce103p, which is involved in CO_2_ hydration and signaling. *C. albicans* lacking Nce103p cannot survive in low CO_2_ concentrations, e.g., in atmospheric growth conditions. *Candida* carbonic anhydrases are orthologous to the *Saccharomyces cerevisiae* enzyme, which had originally been detected as a substrate of a non-classical export pathway. However, experimental evidence on localization of *C. albicans* and *C. parapsilosis* carbonic anhydrases has not been reported to date. Immunogold labeling and electron microscopy used in the present study showed that carbonic anhydrases are localized in the cell wall and plasmatic membrane of both *Candida* species. This localization was confirmed by Western blot and mass spectrometry analyses of isolated cell wall and plasma membrane fractions. Further analysis of *C. albicans* and *C. parapsilosis* subcellular fractions revealed presence of carbonic anhydrases also in the cytosolic and mitochondrial fractions of *Candida* cells cultivated in shaken liquid cultures, under the atmospheric conditions.

## 1. Introduction

The yeasts *Candida albicans* and *Candida parapsilosis* can cause opportunistic infections in humans, especially in hosts whose immune defenses have been weakened. *Candida* species are involved in a range of diseases and are leading causative agents of invasive, life-threatening bloodstream infections. *C. albicans* is responsible for the vast majority of diseases caused by yeasts. *C. parapsilosis* is less virulent than *C. albicans*, but it ranks as the second or third most frequent pathogenic *Candida* species in global epidemiological studies. It often causes infections in low-birth weight infants at neonatal intensive care units [1,2,3].

Both *C. albicans* and *C. parapsilosis* thrive under a wide range of conditions and can metabolically adapt to environmental changes. This metabolic response includes adaptation to changing concentrations of ambient CO_2_. *C. albicans* cannot survive in atmospheric air, which contains approximately 0.04% CO_2_, without expression of the carbonic anhydrase CaNce103p [4,5]. Carbonic anhydrases (CA) catalyze rapid reversible hydration of carbon dioxide into bicarbonate (CO_2_ +H_2_O ↔ HCO_3_^−^ +H^+^). Although this reaction proceeds spontaneously in nature, its catalysis by CAs helps to maintain cellular pH homeostasis and to control a pool of bicarbonate, which is the substrate for metabolic carboxylation reactions. CA (EC 4.2.1.1) thus plays a vital role during the yeast colonization of the host skin or biofilms formation on abiotic surfaces such as plastic medical devices. When *C. albicans* enters the bloodstream, the CO_2_ concentration increases up to 5%, and CA is transcriptionally downregulated [6,7].

Tight control of the CO_2_ fixation reaction is of such general importance that CAs arose in nature independently several times during evolution, yielding structurally different enzymes that are categorized into seven classes: α, β, γ, δ, ζ, η, and θ. The yeast CAs characterized thus far belong to the β-class [8]. CAs from *C. albicans* and *Saccharomyces cerevisiae* have been structurally characterized [9,10] and their indispensability for these yeasts under the atmospheric growth conditions has been demonstrated [4,11].

CA of *S. cerevisiae* was first discovered as one of the proteins that lacked classical secretion signal (SS) and was exported from the yeast cells whose standard secretory pathway has been blocked [12]. The gene name *NCE103* refers to the non-classical protein export pathway. The orthologous genes were identified in *C. albicans* and *C. parapsilosis* are also denominated *NCE103* [4,13]. CAs from *C. albicans* and *C. parapsilosis* will be referred to as CaNce103p and CpNce103p, respectively.

Nce103p enzymes are thought to play a role not only in CO_2_ hydration, but also in CO_2_ sensing and yeast cell signaling. Thus, changing levels of CO_2_ are related to morphology changes in *C. albicans* and contribute to virulence [7]. However, cellular localization of CaNce103p has not been examined. The emerging pathogen *C. parapsilosis* is generally less characterized than *C. albicans*, and CpNce103p has not been experimentally studied at all. In the present study we used immunoelectron microscopy, subcellular fractionation and mass spectrometry, to analyze cellular localization of CAs of two pathogenic *Candida* species.

## 2. Results

### 2.1. Localization of Candida CAs Using Immunogold Labeling and Electron Microscopy

*C. albicans* and *C. parapsilosis* CAs share 72% of identity. The polyclonal antibodies raised against recombinant CpNce103p were found to cross-react with recombinant CaNce103p (Dostál 2019, unpublished results) and were thus instrumental for the localization of CAs in both yeast species studied using electron microscopy and immunogold labeling technique. Figure 1 shows that CAs are predominantly localized in the outer part of the cells, in the cell wall (CW) and plasma membrane (PM) but also in the cytoplasm. The PM localization is visible especially in *C. albicans*. Immunogold labeling detected also populations of Nce103p molecules secreted from *C. albicans* or *C. parapsilosis*.

### 2.2. Mass-Spectrometry Analysis of CW and PM Fractions

To confirm the subcellular localization of CAs, the CW and PM fractions from both *Candida* species were isolated using the protocol of Zinser and Daum [14] and proteins released from these fractions were analyzed by Western blot with anti-CpNce103 antibodies and mass spectrometry (MS). As illustrated by Figure 2A, CAs were detected in the CW fractions of both yeast species. However, only the PM fractions obtained from *C. albicans* displayed a weak signal that may correspond to the CA, which is in an agreement with the EM observations.

MS analysis confirmed the presence of CAs in the PM and CW of both *C. albicans* and *C. parapsilosis* cells. The sequence coverages of PM and CW released proteins using the method that involved SDS, β-mercaptoethanol, and heating were 69% for both CaNce103p and CpNce103p (Figure 2B). The identified peptides in CaNce103p and CpNce103p are summarized in Appendix A, respectively.

### 2.3. Western Blot Analysis of Selected Subcellular Fractions

According to electronic annotation listed in the *Candida* Genome Database, both CaNce103p and CpNce103p were predicted to be localized in the mitochondrial intermembrane space (www.candidagenome.org; 10th December 2019) [13]. In order to examine the presence of CAs in mitochondria and cytosol, mitochondrial and cytosolic fractions were prepared from both *C. albicans* and *C. parapsilosis*, and subjected to the Western blot analysis. The first step in the yeast subcellular fractionation was preparation of spheroplasts, which was based on enzymological removal of the cell wall. Lyticase treatment of the cells yielded the protein fraction (LRP), which was used as an alternatively prepared CW protein fraction and analyzed along with the mitochondria and cytosol. As shown in Figure 3, Western blots for *C. albicans* and *C. parapsilosis.* displayed similar pattern, detecting approximately 16% of total Nce103p proteins in mitochondria of the both yeast species. CAs occurred also in cytosol, but predominantly among the proteins released from the CW using lyticase. Roughly 48% of total CaNce103p and 59% of CpNce103p were detected in the LRP fraction.

Anti-ATP5A antibodies were used as the mitochondrial marker, to confirm the identity of the mitochondrial fraction and to verify absence of mitochondria in the CYT and LRP fractions.

## 3. Discussion

Adaptation to changing CO_2_ concentrations is one of the key and vital abilities of all living cells. *C. albicans* and *C. parapsilosis* regularly occur in the environments, in which the CO_2_ concentrations differ by two orders of magnitude, and require carbonic anhydrases to survive under the atmospheric growth conditions. The gene *NCE103* encoding carbonic anhydrases in both *Candida* species studied received its name on the basis of homology with the *S*. *cerevisiae* gene encoding a non-classical export protein, which was secreted from the cells by a non-classical pathway and lacked a signal peptide [12]. In silico SS prediction using SignalP 4.1 [15] showed the absence of a regular SS in CaNce103p and CpNce103p, similar to their *Saccharomyces* ortholog. In the present work, both CAs were detected in the CW using diverse methodological approaches. CaNce103p and CpNce103p may thus be substrates of the non-classical export pathway in the respective yeast species. According to Cleves et al. [12], non-classical protein export could play a role in removal of proteins that are normally required or tolerated at low expression levels, but become toxic when present at high levels. The present study does not rule out this hypothesis, particularly when CAs were detected also in extracellular space. Nevertheless, the amounts of CAs in CW, as well as in cytosolic and mitochondrial fractions were so low that they had to be immunoprecipitated, in order to enable unambiguous and reproducible detection. *C. albicans* CW proteome has been a subject of many analyses in past [16], and CaNce103p has not occurred among the detected proteins, probably due to its low concentration. Therefore, it seems more likely that CW localization of CAs plays a specific role in ambient CO_2_ sensing and yeast growth. The CO_2_ fixed as HCO_3_^−^ represents important substrate for carboxylation reactions occurring in different metabolic reactions that sustain gluconeogenesis, fatty acid elongation, tricarboxylic acid cycle, replenish C_4_ intermediates, etc. This opens an important question whether the CO_2_ hydrating and potentially also CO_2_ sensing role of CAs is impaired by the antimycotics targeting the CW, such as echinocandins.

Proteomic analyses of mitochondria have not detected CaNce103 either [17], although mitochondrial localization of this CA has been suggested by *Candida* Genome Database, based on the localization of homologous *S. cerevisiae* enzyme in mitochondrial intermembrane space [18]. The present analysis detected CaNce103p as well as CpNce103p in mitochondrial fractions. Mitochondrial localization indicates involvement of the yeast CAs in hydrating intracellular CO_2,_ arising from metabolic processes. While this study investigated the localization of CAs under one set of conditions, it would be interesting to examine whether changes of the carbon source can bring about changes in the CA localization

CaNce103p and CpNce103p have been detected also in cytosol. This is paralleled by the cytosolic localization of CA from *S. cerevisiae* [19]. The question is whether the occurrence of CaNce103p and CpNce103p in cytosol is transient, as they are being transported to other parts of the cells, or if these CAs are ubiquitous enzymes playing different roles at different cellular sites: sensing and hydrating CO_2_ regardless of whether it comes from the environment, or is formed during the metabolic reactions inside the cell.

## 4. Materials and Methods

### 4.1. Strains and Growth Conditions

*C. albicans* strain HE169 and *C. parapsilosis* strain P69 were obtained from the mycological collection of the Faculty of Medicine, Palacky University, Olomouc, Czech Republic. The yeasts were streaked on YPD (2% peptone, 1% yeast extract, 2% glucose) agar and incubated at 30 °C for 24 h. Single colonies were picked to inoculate 10 mL of YPD as a preculture. An aliquot (200 μL) of the stationary phase preculture was then inoculated into 500 mL of YPD medium and cultivated in the rotation shaker at 250 rpm and 30 °C.

### 4.2. Immunogold Labeling and Electron Microscopy

Immunogold labeling was performed using rabbit polyclonal antibodies (Moravian-Biotechnology, Brno, Czech Republic), which were raised against recombinant CpNce103p expressed in *Escherichia coli* and purified, as described by Dostál et al. [10]. These antibodies interacted with both CpNce103p and CaNce103p. Samples for EM analysis were prepared by cultivation of the yeasts in YPD at 30 °C overnight. Cells were harvested by centrifugation (5000× *g* for 15 min), and mixed with dextran in PBS buffer pH 7.4 up to its final concentration of 20%. The samples were immediately fast frozen by Leica EM PACT2 High Pressure Freezer (Leica-Microsystems, Vienna, Austria). Freeze substitution was carried out using a Leica EM AFS apparatus (Leica-Microsystems, Vienna, Austria) in dry acetone containing 0.5% uranyl acetate for 8 h and sequences of −90, −70, and −50 °C. The samples were washed, infiltrated into Lowicryl HM20 resin and polymerized with UV radiation. Ultrathin sections (70 nm) prepared on ultramicrotome (Leica Ultracent EM UC7, Leica-Microsystems, Vienna, Austria)) were mounted on parlodion-coated nickel grids, which were subsequently blocked for 25 minutes using the buffer consisting of 1% BSA (*w/v*) in PBS, pH 7.4 and 10% normal goat serum (*v/v*) and incubated in buffer A (0.5% BSA (*w/v*), 0.05% TWEEN 20 (*v/v*) in PBS, pH 7.4) containing anti-CpNce103p antibody diluted 1:100, overnight at 4 °C. After washing with buffer A, the grids were treated with a droplet of goat anti-rabbit IgG conjugated to 10 nm gold particles (British Biocell, Cardiff, UK), diluted 1:25 in buffer A, and incubated for 1 h 4 °C. Then, the grids were washed, stained with uranyl acetate, and viewed with a JEOL JEM 1011 electron microscope at 80 kV (JEOL Europe SAS, Croissy sur Seine, France).

### 4.3. Isolation of the Cell Wall

Cell wall (CW) and plasma membrane (PM) fractions were prepared based on the protocols described by Zinser and Daum and Pitarch et al. [14,20]. Brifly, *Candida* cells were harvested by centrifugation at 4000× *g* for 10 minutes, resuspended in 30 mM TRIS, 50 mM ammonium acetate, pH 8.5, supplemented with 5 mM EDTA and 0.5 mM PefaBloc (Sigma-Aldrich, St. Louis, MO, USA), disintegrated using EmulsiFlex-C3 homogenizer (Biopharma Group, Dublin, Republic of Ireland) and brief sonication, and centrifuged at 2000× *g* for 10 minutes. The sediment containing the crude CW was washed five times with each of the following solutions: ice-cold 5% NaCl, 2% NaCl, 1% NaCl, and water. Supernatant was further centrifuged at 20,000× *g* for 20 min, yielding PM in sediment. CW and PM were resuspended in 50 mM Tris-HCl, pH 7.5 containing 2% SDS, 0.1 M EDTA, 40 mM β-mercaptoethanol. To release the proteins, the suspensions were heated to 100 °C for 5 min and then centrifuged (5 min 2000× *g*). The supernatants were dialyzed against 50 mM Tris-Cl, 150 mM NaCl, pH 7.5 [21].

### 4.4. Mass Spectrometry Analysis of Nce103p in the Cell Wall Samples

CAs potentially localized in CW and PM were immunoprecipitated using polyclonal anti-CpNce103p antibodies bound to Protein A Mag Sepharose^TM^ beads (GE Healthcare) using the manufacturer’s procedure. A negative control without antibodies was also prepared. The antibody-cross-linked beads were incubated overnight at 4 °C with 600 µL solution of proteins released from PM and CW. After incubation, the immunoprecipitated proteins were washed with TBS buffer (50 mM Tris-Cl, 150 mM NaCl pH 7.5) five times, and eluted with 50 µL of 0.1 M glycine-HCl, pH 2.9, containing 2 M urea and subjected to Western blotting using polyclonal antibodies against CpNce103p and mass-spectrometry (MS) analysis. Proteins for MS were either reduced on the magnetic beads by dithiothreitol and alkylated by iodoacetamide or left in their native form and then digested by trypsin overnight at pH 8.5. Desalted peptides eluted from Acclaim PepMap100 column were analyzed using UltiMate 3000 RSLCnano system (Thermo Fisher Scientific, Waltham, MA, USA 02451) coupled to Orbitrap Fusion Lumos (Thermo Fisher Scientific, MA, USA, 02451) using orbitrap in both MS and MSMS acquisition steps in range 100–2000 *m*/*z*. Protein identification was performed using the UniProt sequence database (CpNce103p, UniProtKB: G8B6R8 and CaNce130p, UniProtKB:Q5AJ71) with Proteome discoverer 2.3 software (Thermo Fisher Scientific, MA, USA).

### 4.5. Preparation of Spheroplasts and Subcellular Fractions

Mid-exponential phase *Candida* cells were harvested from YPD cultures by centrifugation at 4000× *g* for 10 min. The cells were washed with ice-cold 10 mM NaN_3_, resuspended in SPB buffer (0.8 M sorbitol, 40 mM β-mercaptoethanol, 50 mM KH_2_PO_4_, pH 7.5). The suspension was treated with Lyticase (Sigma-Aldrich, St. Louis, MO, USA) at a concentration of 1 mg/100 mg wet biomass. The mixture was incubated at 30 °C with shaking at 250 rpm until the OD_600_ of the suspension dropped to at least 10% of its initial value. Then, the suspension was centrifuged at 1500× *g* for 15 min at 4 °C. Supernatant was used as lyticase-released cell wall proteins (LRP), and sediment containing spheroplasts was used for subcellular fractionation.

The harvested *Candida* spheroplasts were washed twice in 1 M sorbitol and resuspended in ice-cold SPB buffer, supplemented with 1 mM EDTA. Disintegration was performed by using Dounce homogenizer. The lysate was centrifuged at 2500× *g* and 4 °C in order to remove unbroken or partially disrupted cells and aggregates. The resulting supernatant was further centrifuged at 10,000× *g* for 30 min, 4 °C. The supernatant was used as cytosolic fraction (CYT). The pellet was resuspended in 2 mL of buffer A (250 mM sucrose, 1 mM EDTA, 10 mM HEPES pH7.4) and layered on the top of a 10 mM HEPES-buffered sucrose step gradient (2 mL, 15%; 3 mL, 25%; 3 mL, 40%; and 2 mL, 60%, respectively) and centrifuged for 1 h at 4 °C at 100,000× *g* in a SW41Ti rotor (Beckman Coulter, Brea, CA, USA). A reddish-brown mitochondria-enriched layer was collected and used as a mitochondrial fraction (MIT).

### 4.6. Western Blot Analysis of Subcellular Fractions

MIT fraction (50 mg of protein dissolved in 50 µL PLB buffer) was subjected to Western blotting analysis directly. LRP and CYT fractions were immunoprecipitated using anti-CpNce103 polyclonal antibodies. In a parallel experiment, proteins were immunoprecipitated using antibodies against peptide derived from the mitochondrial ATP synthase (anti-ATP5A; Abcam, Cambridge, UK), to provide a control of potential contamination of LRP and CYT by mitochondria. The antibodies were bound to Protein A Mag Sepharose TM (GE Healthcare, Chicago, IL, USA) beads using the manufacturer’s procedure. The negative control without antibodies was prepared similarly. The ready antibody-cross-linked beads were incubated overnight at 4 °C with 5 mL of CYT and LRP, in which the protein concentration was adjusted approximately to 10 mg/mL. After incubation, the beads with immunoprecipitated proteins were washed with TBS buffer (50 mM Tris-Cl, 150 mM NaCl pH 7.5) five times and resuspended in 50 µL of PLB buffer (4% SDS, 10% 2-mercaptoethanol, 20% glycerol, 0.004% bromophenol blue, 0.125 M Tris-HCl pH 6.8). Analyzed proteins were separated on 12% SDS-PAGE gel, blotted on nitrocellulose membrane and detected by using polyclonal anti-CpNce103p or anti-ATP5A antibodies (Abcam, Cambridge, UK). Peroxidase-labeled swine anti-rabbit immunoglobulins were used as the secondary antibodies, and the visualization was performed using the West Femto detection system (Thermo Fisher Scientific, Waltham, MA, USA). Band intensities for CaNce103p and CpNce103p were analyzed using a Typhoon system and ImageQuant software (Amersham Biosciences, version TL 8.1, Little Chalfont, UK).

## 5. Conclusions

We used several methods to demonstrate that the Nce103p CAs of *C. albicans* and *C. parapsilosis* are predominantly localized in the cell wall and plasma membrane but partially also in the cytoplasm and mitochondria, suggesting important role of CAs in CO_2_ sensing and regulation of bicarbonate level for intracellular carboxylation reactions.

## Figures and Tables

**Figure 1 ijms-21-00850-f001:**
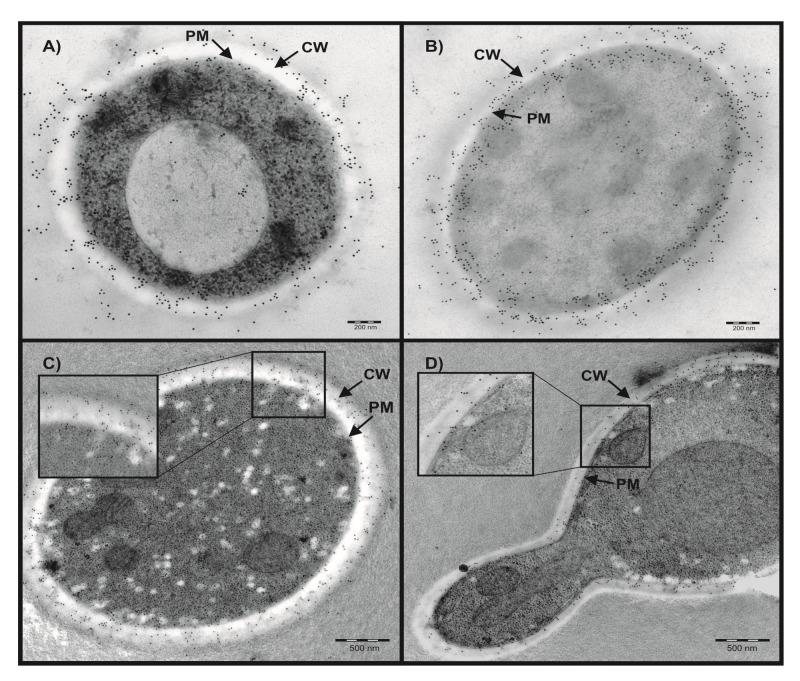
Localization of Nce103p using immunogold labeling and electron microscopy. (**A**,**C**) *C. parapsilosis*; (**B**,**D**) *C. albicans*; CW: cell wall, PM: plasma membrane. Scale bars, 200 nm and 500 nm. Panels C and D contain also zoom-in sections of the outer parts of the cells.

**Figure 2 ijms-21-00850-f002:**
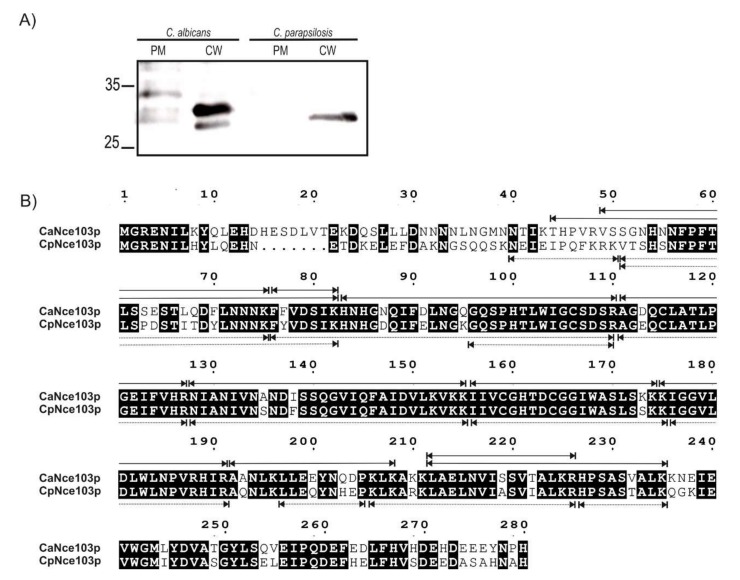
Analysis of Nce103p presence in PM and CW fractions of *C. albicans and C. parapsilosis*. (**A**) Western blot analysis of PM and CW fractions obtained using the protocol of Zinser and Daum [14]. CAs were detected using polyclonal rabbit antibodies against CpNce103p, diluted 1:1000. (**B**) Mass spectrometry (MS) analysis of CaNce103p and CpNce103p in *C. albicans and C. parapsilosis* CWs. The conserved amino acids are highlighted in black. Peptides detected by MS analysis are marked with arrows above (CaNce103p) or below (CpNce103p) sequences (CpNce103p, UniProtKB: G8B6R8 and CaNce130p, UniProtKB:Q5AJ71).

**Figure 3 ijms-21-00850-f003:**
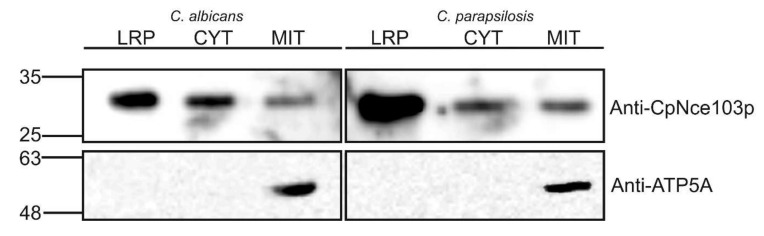
Western blot analysis of CA presence in *C. albicans* and *C. parapsilosis* mitochondrial (MIT) and cytosolic fractions (CYT), and in the fraction obtained by the lyticase treatment of the cell was (LRP). Polyclonal anti-CpNce103p antibodies (upper panel) and anti-ATP5A antibodies (lower panel) were used for the detection.

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
