# Peer review of "Cellular Localization of Carbonic Anhydrase Nce103p in Candida albicans and Candida parapsilosis"

_ijms, 2020, doi:10.3390/ijms21030850_

Round 1
Reviewer 1 Report
The authors report on the experimental localization of ß-type carbonic anhydrases on C. albicans and C. parapsilosis using several methodological approaches.
The introduction is well written and provides relevant information about the state of the art.
Regarding the experimental section, the authors could improve the image quality of the western plot (Fig 2A).
The MS data for the peptide sequences presented in Fig 2B is missing.
On the discussion, the authors should comment on the amounts of CA in each of the discussed compartments and possibly quantify the expression in the compartment with the highest expression.
The sentence that starts in line 218 is not clear.
Author Response
Responses to reviewer 1
The authors report on the experimental localization of ß-type carbonic anhydrases on C. albicans and C. parapsilosis using several methodological approaches.
The introduction is well written and provides relevant information about the state of the art.
Regarding the experimental section, the authors could improve the image quality of the western plot (Fig 2A).
We have improved the quality of Figure 2A
The MS data for the peptide sequences presented in Fig 2B is missing.
We have included supplementary table S1 and S2 containing the MS data.
On the discussion, the authors should comment on the amounts of CA in each of the discussed compartments and possibly quantify the expression in the compartment with the highest expression.
We have detected the CAs levels in individual cell fractions, added the text in methods at page 4 ( Band intensities for CaNce103p and CpNce103p were analyzed using a Typhoon system and ImageQuant software (Amersham). New text describing the CAs levels in the cellular fractions is present at page 6: Western blots for C. albicans and C. parapsilosis displayed similar pattern, showing approximately 16% of total Nce103p proteins in mitochondria of the both yeast species. CAs occurred also in cytosol, but predominantly among the proteins released from the CW using lyticase. Roughly 48% of total CaNce103p and 59% of CpNce103p were detected in the LRP fraction.
The sentence that starts in line 218 is not clear.
There is a new text here: CAs plays a specific role in ambient CO2 sensing and yeast growth. The CO2 fixation in a form of HCO3− is important for maintaining of different metabolic reactions that sustain gluconeogenesis, replenishment of C4 intermediates, fatty acid elongation, tricarboxylic acid cycle, etc. This opens and important question whether the CO2 hydrating and potentially also CO2 sensing role of CAs is impaired by the antimycotics targeting the CW, such as echinocandins.

Reviewer 2 Report
Dostal et al. in the paper entitled “Cellular localization of carbonic anhydrase Nec103p in C. albicans and C. parapsilosis” show for the first time the cellular localization of carbonic anhydrase (CA) experimentally. They showed using immunogold labelling and electron microscopy that CA is localized in plasma membrane and in cell wall in both species. Further, they confirmed the localization of CA in cell wall and plasma membrane using western blot and mass spectroscopy analysis of isolated cell wall and plasma membrane fractions. In addition, they show that CA is also localized in cytosol and mitochondria of both species when the cells are cultivated in shaken liquid cultures, under atmospheric conditions.
The experiments have been carefully performed and are technically sound. Their results support the conclusion. I recommend publication of this article in IJMS provided the authors address the following:
Minor Comments:
What is the significance of CAs localization in Cell wall, Plasma membrane, cytosol, and Mitochondria? Do CAs localizes in different compartments play different roles in signaling/ colonization? Do the authors think that the drugs that are used to treat the diseases caused by these Candida’s somehow chase the localization of CAs in them? There are some typos in result section.Author Response
Minor Comments:
What is the significance of CAs localization in Cell wall, Plasma membrane, cytosol, and Mitochondria? Do CAs localizes in different compartments play different roles in signaling/ colonization? Do the authors think that the drugs that are used to treat the diseases caused by these Candida’s somehow chase the localization of CAs in them? There are some typos in result section.
We have added a new text in the discussion:
“It therefore seems more likely that CW localization of CAs plays a specific role in ambient CO2 sensing and yeast growth. The CO2 fixation in a form of HCO3− is important for maintaining of different metabolic reactions that sustain gluconeogenesis, replenishment of C4 intermediates, fatty acid elongation, tricarboxylic acid cycle, etc. This opens and important question whether the CO2 hydrating and potentially also CO2 sensing role of CAs is impaired by the antimycotics targeting the CW, such as echinocandins„

Round 2
Reviewer 2 Report
The manuscript has improved after revision. It has also addressed my comments.